# Faster Width-dependent Algorithm for Mixed Packing and Covering LPs

**Digvijay Boob**
Georgia Tech
Atlanta, GA
digvijaybb40@gatech.edu

**Saurabh Sawlani**
Georgia Tech
Atlanta, GA
sawlani@gatech.edu

**Di Wang***
Google AI
Atlanta, GA
wadi@google.com

## Abstract

In this paper, we give a faster width-dependent algorithm for mixed packing-covering LPs. Mixed packing-covering LPs are fundamental to combinatorial optimization in computer science and operations research. Our algorithm finds a $1 + \varepsilon$ approximate solution in time $O(Nw/\varepsilon)$, where $N$ is number of nonzero entries in the constraint matrix, and $w$ is the maximum number of nonzeros in any constraint. This algorithm is faster than Nesterov's smoothing algorithm which requires $O(N\sqrt{n}w/\varepsilon)$ time, where $n$ is the dimension of the problem. Our work utilizes the framework of area convexity introduced in [Sherman-FOCS'17] to obtain the best dependence on $\varepsilon$ while breaking the infamous $\ell_\infty$ barrier to eliminate the factor of $\sqrt{n}$. The current best width-independent algorithm for this problem runs in time $O(N/\varepsilon^2)$ [Young-arXiv-14] and hence has worse running time dependence on $\varepsilon$. Many real life instances of mixed packing-covering problems exhibit small width and for such cases, our algorithm can report higher precision results when compared to width-independent algorithms. As a special case of our result, we report a $1 + \varepsilon$ approximation algorithm for the densest subgraph problem which runs in time $O(md/\varepsilon)$, where $m$ is the number of edges in the graph and $d$ is the maximum graph degree.

## 1 Introduction

Mixed packing and covering linear programs (LPs) are a natural class of LPs where coefficients, variables, and constraints are non-negative. They model a wide range of important problems in combinatorial optimization and operations research. In general, they model any problem which contains a limited set of available resources (packing constraints) and a set of demands to fulfill (covering constraints).

Two special cases of the problem have been widely studied in literature: pure *packing*, formulated as $\max_x\{b^T x \mid Px \leqslant p\}$; and pure *covering*, formulated as $\min_x\{b^T x \mid Cx \geqslant c\}$ where $P, p, C, c, b$ are all non-negative. These are known to model fundamental problems such as maximum bipartite graph matching, minimum set cover, etc. [LN93]. Algorithms to solve packing and covering LPs have also been applied to great effect in designing flow control systems [BBR04], scheduling problems [PST95], zero-sum matrix games [Nes05] and in mechanism design [ZN01]. In this paper, we study the mixed packing and covering (MPC) problem, formulated as checking the feasibility of the set: $\{x \mid Px \leqslant p, Cx \geqslant c\}$, where $P, C, p, c$ are non-negative. We say that $x$ is an $\varepsilon$-approximate solution to MPC if it belongs to the relaxed set $\{x \mid Px \leqslant (1+\varepsilon)p, Cx \geqslant (1-\varepsilon)c\}$. MPC is a generalization of pure packing and pure covering, hence it is applicable to a wider range of problems such as multi-commodity flow on graphs [You01, She17], non-negative linear systems and X-ray tomography [You01].

General LP solving techniques such as the interior point method can approximate solutions to MPC in as few as $O(\log(1/\varepsilon))$ iterations - however, they incur a large per-iteration cost. In contrast, iterative approximation algorithms based on first-order optimization methods require $\text{poly}(1/\varepsilon)$ iterations, but the iterations are fast and in most cases are conducive to efficient parallelization. This property is of utmost importance in the context of ever-growing datasets and the availability of powerful parallel computers, resulting in much faster algorithms in relatively low-precision regimes.

## 1.1 Previous work

In literature, algorithms for the MPC problem can be grouped into two broad categories: *width-dependent* and *width-independent*. Here, *width* is an intrinsic property of a linear program which typically depends on the dimensions and the largest entry of the constraint matrix, and is an indication of the range of values any constraint can take. In the context of this paper and the MPC problem, we define $w_P$ and $w_C$ as the maximum number of non-zeros in any constraint in $P$ and $C$ respectively. We define the width of the LP as $w \stackrel{\text{def}}{=} \max(w_P, w_C)$.

One of the first approaches used to solve LPs was Langrangian-relaxation: replacing hard constraints with loss functions which enforce the same constraints indirectly. Using this approach, Plotkin, Schmoys and Tardos [PST95], and Grigoriadis and Khachiyan [GK96] obtained width-dependent polynomial-time approximation algorithms for MPC. Luby and Nisan [LN93] gave the first width-dependent parallelizable algorithm for pure packing and pure covering, which ran in $\widetilde{O}(\varepsilon^{-4})$ parallel time, and $\widetilde{O}(N\varepsilon^{-4})$ total work. Here, *parallel time* (sometimes termed as *depth*) refers to the longest chain of dependent operations, and *work* refers to the total number of operations in the algorithm.

Young [You01] extended this technique to give the first width-independent parallel algorithm for MPC in $\widetilde{O}(\varepsilon^{-4})$ parallel time, and $\widetilde{O}(md\varepsilon^{-2})$ total work[2]. Young [You14] later improved his algorithm to run using total work $O(N\varepsilon^{-2})$. Mahoney *et al*. [MRWZ16] later gave an algorithm with a faster parallel run-time of $\widetilde{O}(\varepsilon^{-3})$.

The other most prominent approach in literature towards solving an LP is by converting it into a smooth function [Nes05], and then applying general first-order optimization techniques [Nes05, Nes12]. Although the dependence on $\varepsilon$ from using first-order techniques is much improved, it usually comes at the cost of sub-optimal dependence on the input size and width. For the MPC problem, Nesterov's accelerated method [Nes12], as well as Bienstock and Iyengar's adaptation [BI06] of Nesterov's smoothing [Nes05], give rise to algorithms with runtime linearly depending on $\varepsilon^{-1}$, but with far from optimal dependence on input size and width. For pure packing and pure covering problems, however, Allen-Zhu and Orrechia [AO19] were the first to incorporate Nesterov-like acceleration while still being able to obtain near-linear width-independent runtimes, giving a $\widetilde{O}(N\varepsilon^{-1})$ time algorithm for the packing problem. For the covering problem, they gave a $\widetilde{O}(N\varepsilon^{-1.5})$ time algorithm, which was then improved to $\widetilde{O}(N\varepsilon^{-1})$ by [WRM16]. Importantly, however, the above algorithms do not generalize to MPC.

## 1.2 Our contributions

We give the best parallel width-dependent algorithm for MPC, while only incurring a linear dependence on $\varepsilon^{-1}$ in the parallel runtime and total work. Additionally, the total work has near-linear dependence on the input-size. Formally, we state our main theorem as follows.

**Theorem 1.1.** *There exists a parallel $\varepsilon$-approximation algorithm for the mixed packing covering problem, which runs in $\widetilde{O}(w \cdot \varepsilon^{-1})$ parallel time, while performing $\widetilde{O}(w \cdot N \cdot \varepsilon^{-1})$ total work, where $N$ is the total number of non-zeros in the constraint matrices, and $w$ is the width of the given LP.*

Table 1 compares the running time of our algorithm to previous works solving this problem.

Sacrificing width independence for faster convergence with respect to precision proves to be a valuable trade-off for several combinatorial optimization problems which naturally have a low width. Prominent examples of such problems which are not pure packing or covering problems include *multicommodity flow* and *densest subgraph*, where the width is bounded by the degree of a vertex. In a large number of real-world graphs, the maximum vertex degree is usually small, hence our

Table 1: Comparison of runtimes of $\varepsilon$-approximation algorithms for the mixed packing covering problem.

| | Parallel Runtime | Total Work | Comments |
|---|---|---|---|
| Young [You01] | $\widetilde{O}(\varepsilon^{-4})$ | $\widetilde{O}(md\varepsilon^{-2})$ | $d$ is column-width |
| Bienstock and Iyengar [BI06] | | $\widetilde{O}(n^{2.5}w_P^{1.5}w\varepsilon^{-1})$ | width-dependent |
| Nesterov [Nes12] | $\widetilde{O}(w\sqrt{n}\varepsilon^{-1})$ | $\widetilde{O}(w \cdot N\sqrt{n}\varepsilon^{-1})$ | width-dependent |
| Young [You14] | $\widetilde{O}(\varepsilon^{-4})$ | $\widetilde{O}(N\varepsilon^{-2})$ | |
| Mahoney *et al.* [MRWZ16] | $\widetilde{O}(\varepsilon^{-3})$ | $\widetilde{O}(N\varepsilon^{-3})$ | |
| This paper | $\widetilde{O}(w\varepsilon^{-1})$ | $\widetilde{O}(wN\varepsilon^{-1})$ | width-dependent |

algorithm proves to be much faster when we want high-precision solutions. We explicitly show that this result directly gives the fastest algorithm for the densest subgraph problem on low-degree graphs in Appendix C.

## 2   Notation and Definitions

For any integer $q$, we represent using $\|\cdot\|_q$ the $q$-norm of any vector. We represent the infinity-norm as $\|\cdot\|_\infty$. We denote the infinity-norm ball (sometimes called the $\ell_\infty$ ball) as the set $\mathcal{B}_\infty^n(r) \stackrel{\text{def}}{=} \{x \in \mathbb{R}^n : \|x\|_\infty \leqslant r\}$. The nonnegative part of this ball is denoted as $\mathcal{B}_{+,\infty}^n(r) = \{x \in \mathbb{R}^n : x \geqslant \mathbf{0}_n, \|x\|_\infty \leqslant r\}$. For radius $r = 1$, we drop the radius specification and use the short notation $\mathcal{B}_\infty^n$ and $\mathcal{B}_{+,\infty}^n$. We denote the extended simplex of dimension $k$ as $\Delta_k^+ \stackrel{\text{def}}{=} \{x \in \mathbb{R}^k : \sum_{i=1}^k x_i \leqslant 1\}$. For any $y \geqslant \mathbf{0}_k$, $\text{proj}_{\Delta_k^+}(y) = y/\|y\|_1$ if $\|y\|_1 \geqslant 1$. Further, for any set $K$, we represent its interior, relative interior and closure as $\text{int}(K), \text{relint}(K)$ and $\text{cl}(K)$, respectively. The function $\exp$ is applied to a vector element wise. The division of two vectors of same dimension is also performed element wise.

For any matrix $A$, we use $\text{nnz}(A)$ to denote the number of nonzero entries in it. We use $A_{i,:}$ and $A_{:,j}$ to refer to the $i$th row and $j$th column of $A$ respectively. We use notation $A_{ij}$ (or $A_{i,j}$ alternatively) to denote an element in the $i$-th row and $j$-th column of matrix $A$. $\|A\|_\infty$ denotes the operator norm $\|A\|_{\infty\to\infty} \stackrel{\text{def}}{=} \sup_{x\neq 0} \frac{\|Ax\|_\infty}{\|x\|_\infty}$. For a symmetric matrix $A$ and an antisymmetric matrix $B$, we define an operator $\succeq_\text{i}$ as $A \succeq_\text{i} B \Leftrightarrow \begin{bmatrix} A & -B \\ B & A \end{bmatrix}$ is positive semi-definite.

We formally define an $\varepsilon$-approximate solution to the mixed packing-covering (MPC) problem as follows.

**Definition 2.1.** We say that $x$ is an $\varepsilon$-approximate solution of the mixed packing-covering problem if $x$ satisfies $x \in \mathcal{B}_{+,\infty}^n$, $Px \leqslant (1+\varepsilon)\mathbf{1}_p$ and $Cx \geqslant (1-\varepsilon)\mathbf{1}_c$.

Here, $\mathbf{1}_k$ denotes a vectors of 1's of dimension $k$ for any integer $k$.

The saddle point problem on two sets $x \in X$ and $y \in Y$ can be defined as follows:

$$\min_{x\in X} \max_{y\in Y} \mathcal{L}(x,y), \tag{1}$$

where $\mathcal{L}(x,y)$ is some bilinear form between $x$ and $y$. For this problem, we define the *primal-dual gap function* as $\sup_{(\bar{x},\bar{y})\in X\times Y} \mathcal{L}(x,\bar{y}) - \mathcal{L}(\bar{x},y)$. This gap function can be used as measure of accuracy of the above saddle point solution.

**Definition 2.2.** We say that $(x,y) \in X \times Y$ is an $\varepsilon$-optimal solution for (1) if $\sup_{(\bar{x},\bar{y})\in X\times Y} \mathcal{L}(x,\bar{y}) - \mathcal{L}(\bar{x},y) \leqslant \varepsilon$.

## 3   Technical overview

The mixed packing-covering (MPC) problem is formally defined as follows.

> Given two nonnegative matrices $P \in \mathbb{R}^{p\times n}$, $C \in \mathbb{R}^{c\times n}$, find an $x \in \mathbb{R}^n, x \geqslant \mathbf{0}, \|x\|_\infty \leqslant 1$ such that $Px \leqslant \mathbf{1}_p$ and $Cx \geqslant \mathbf{1}_c$ if it exists, otherwise report infeasibility.

Note that the vector of 1's on the right hand side of the packing and covering constraints can be obtained by simply scaling each constraint appropriately. We also assume that each entry in the matrices $P$ and $C$ is at most 1. This assumption, and subsequently the $\ell_\infty$ constraints on $x$ also cause no loss of generality[3].

We reformulate MPC as a saddle point problem, as defined in Section 2;

$$\lambda* \stackrel{\text{def}}{=} \min_{x \in \mathcal{B}^n_{+,\infty}} \max_{y \in \Delta^+_c, \ z \in \Delta^+_p} L(x, y, z), \tag{2}$$

where $L(x, y, z) := \begin{bmatrix} y^T & z^T \end{bmatrix} \begin{bmatrix} P & -\mathbf{1}_p \\ -C & \mathbf{1}_c \end{bmatrix} \begin{bmatrix} x \\ 1 \end{bmatrix}$. The relation between the two formulations is shown in Section 4. For the rest of the paper, we focus on the saddle point formulation (2).

$\eta(x) \stackrel{\text{def}}{=} \max_{y \in \Delta^+_c, z \in \Delta^+_p} L(x, y, z)$ is a piecewise linear convex function. Assuming oracle access to this "inner" maximization problem, the "outer" problem of minimizing $\eta(x)$ can be performed using first order methods like mirror descent, which are suitable when the underlying problem space is the unit $\ell_\infty$ ball. One drawback of this class of methods is that their rate of convergence, which is standard for non-accelerated first order methods on non-differentiable objectives, is $O(\frac{1}{\varepsilon^2})$ to obtain an $\varepsilon$-approximate minimizer $x$ of $\eta$ which satisfies $\eta(x) \leq \eta* + \varepsilon$, where $\eta*$ is the optimal value. This means that the algorithm needs to access the inner maximization oracle $O(\frac{1}{\varepsilon^2})$ times, which can become prohibitively large in the high precision regime.

Note that even though $\eta$ is a piecewise linear non-differentiable function, it is not a black box function, but a maximization linear functions in $x$. This structure can be exploited using Nesterov's smoothing technique [Nes05]. In particular, $\eta(x)$ can be approximated by choosing a strongly convex[3] function $\phi : \Delta^+_p \times \Delta^+_c \to \mathbb{R}$ and considering

$$\widetilde{\eta}(x) = \max_{y \in \Delta^+_c, z \in \Delta^+_p} L(x, y, z) - \phi(y, z).$$

This strongly convex regularization yields that $\widetilde{\eta}$ is a Lipschitz-smooth[4] convex function. If $L$ is the constant of Lipschitz smoothness of $\widetilde{\eta}$ then application of any of the accelerated gradient methods in literature will converge in $O(\sqrt{L/\varepsilon})$ iterations. Moreover, it can also be shown that in order to construct a smooth $\varepsilon$-approximation $\widetilde{\eta}$ of $\eta$, the Lipschitz smoothness constant $L$ can be chosen to be of the order $O(1/\varepsilon)$, which in turn implies an overall convergence rate of $O(1/\varepsilon)$. In particular, Nesterov's smoothing achieves an oracle complexity of $O((\|P\|_\infty + \|C\|_\infty) D_x \max\{D_y, D_z\} \varepsilon^{-1})$, where where $D_x$, $D_y$ and $D_z$ denote the sizes of the ranges of their respective regularizers which are strongly convex functions. $D_y$ and $D_z$ can be made of the order of $\log p$ and $\log c$, respectively. However, $D_x$ can be problematic since $x$ belongs to an $\ell_\infty$ ball. More on this will soon follow.

Nesterov's dual extrapolation algorithm[Nes07] gives a very similar complexity but is a different algorithm in that it directly addresses the saddle point formulation (2) rather than viewing the problem as optimizing a non-smooth function $\eta$. The final convergence for the dual extrapolation algorithm is given in terms of the *primal-dual gap* function of the saddle point problem (2). This algorithms views the saddle point problem as solving variational inequality for an appropriate monotone operator in joint domain $(x, y, z)$. Moreover, as opposed to smoothing techniques which only regularize the dual, this algorithm regularizes both primal and dual parts *(joint regularization)*, hence is a different scheme altogether.

Note that for both schemes mentioned above, the maximization oracle itself has an analytical expression which involves matrix-vector multiplication. Hence each call to the oracle incurs a sequential run-time of $\text{nnz}(P) + \text{nnz}(C)$. Then, overall complexity for both schemes is of order $O((\text{nnz}(P) + \text{nnz}(C))(\|P\|_\infty + \|C\|_\infty) D_x \max\{D_y, D_z\} \varepsilon^{-1})$.

**The $\ell_\infty$ barrier**

Note that the first method, i.e., Nesterov's smoothing technique has known lower bounds due to [GN15] (see Corollary 1 in their paper). According to their result, the framework of Nesterov's smoothing has a known limitation since it only regularizes the dual variables. As opposed to this, Nesterov's dual extrapolation regularizes both primal and dual variables, and has potential to skip the earlier mentioned lower bounds of [GN15]. However, the complexity result of this method involves a $D_x$ term, which denotes the range of a convex function over the domain of $x$. The following lemma states a lower bound for this range in case of $\ell_\infty$ balls.

**Lemma 3.1.** *Any strongly convex function has a range of at least $\Omega(\sqrt{n})$ on any $\ell_\infty$ ball.*

Since $D_x = \Omega(\sqrt{n})$ for each member function of this wide class, there is no hope of eliminating this $\sqrt{n}$ factor using techniques involving explicit use of strong convexity.

So, the goal now is to find a joint regularization function with a small range over $\ell_\infty$ balls, but still act as good enough regularizers to enable accelerated convergence of the descent algorithm. In pursuit of breaking this $\ell_\infty$ barrier, we draw inspiration from the notion of *area convexity* introduced by Sherman [She17]. Area convexity is a weaker notion than strong convexity, however, it is still strong enough to ensure that accelerated first order methods still go through when using area convex regularizers. Since this is a weaker notion than strong convexity, we can construct area convex functions which have range of $O(n^{o(1)})$ on $\ell_\infty$ ball.

First, we define area convexity, and then go on to mention its relevance to the saddle point problem (2).

Area convexity is a notion defined in context of a matrix $A \in \mathbb{R}^{a \times b}$ and a convex set $K \subseteq \mathbb{R}^{a+b}$. Let $M_A \stackrel{\text{def}}{=} \begin{bmatrix} \mathbf{0}_{b \times b} & -A^T \\ A & \mathbf{0}_{a \times a} \end{bmatrix}$.

**Definition 3.2** ([She17])**.** A function $\phi$ is *area convex* with respect to a matrix $A$ on a convex set $K$ iff for any $t, u, v \in K$, $\phi$ satisfies $\phi\left(\frac{t+u+v}{3}\right) \leqslant \frac{1}{3}\left(\phi(t) + \phi(u) + \phi(v)\right) - \frac{1}{3\sqrt{3}}(v-u)^T M_A(u-t)$.

To understand the definition above, let us first look at the notion of strong convexity. $\phi$ is said to be strongly convex if for any two points $t, u$, $\frac{1}{2}(\phi(t) + \phi(u))$ exceeds $\phi(\frac{1}{2}(t+u))$ by an amount proportional to $\|t - u\|_2^2$. Definition 3.2 generalizes this notion in context of matrix $A$ for any three points $x, y, z$. $\phi$ is area-convex on set $K$ if for any three points $t, u, v \in K$, we have $\frac{1}{3}(\phi(t) + \phi(u) + \phi(v))$ exceeds $\phi(\frac{1}{3}(t + u + v))$ by an amount proportional to the area of the triangle defined by the convex hull of $t, u, v$.

Consider the case that points $t, u, v$ are collinear. For this case, the area term (i.e., the term involving $M_A$) in Definition 3.2 is 0 since matrix $M_A$ is antisymmetric. In this sense, area convexity is even weaker than strict convexity. Moreover, the notion of area is parameterized by matrix $A$. To see a specific example of this notion of area, consider $A = \begin{bmatrix} 0 & -1 \\ 1 & 0 \end{bmatrix}$ and $t, u, v \in \mathbb{R}^2$. Then, for all possible permutations of $t, u, v$, the area term takes a value equal to $\pm(t_1(u_2 - v_2) + u_1(v_2 - t_2) + v_1(t_2 - u_2))$. Since the condition holds irrespective of the permutation so we must have that $\phi(\frac{t+u+v}{3}) \leqslant \frac{1}{3}\left(\phi(t) + \phi(u) + \phi(v)\right) - \frac{1}{3\sqrt{3}}|t_1(u_2 - v_2) + u_1(v_2 - t_2) + v_1(t_2 - u_2)|$. But note that area of triangle formed by points $t, u, v$ is equal to $\frac{1}{2}|t_1(u_2 - v_2) + u_1(v_2 - t_2) + v_1(t_2 - u_2)|$. Hence the area term is just a high dimensional matrix based generalization of the area of a triangle.

Coming back to the saddle point problem (2), we need to pick a suitable area convex function $\phi$ on the set $\mathcal{B}_{+,\infty}^n \times \Delta_p^+ \times \Delta_c^+$. Since $\phi$ is defined on the joint space, it has the property of joint regularization vis a vis (2). However, we need an additional parameter: a suitable matrix $M_A$. The choice of this matrix is related to the bilinear form of the *primal-dual gap function* of (2). We delve into the technical details of this in Section 4, however, we state that the matrix is composed of $P, C$ and some additional constants. The algorithm we state exactly follows Nesterov's dual extrapolation method described earlier. One notable difference is that in [Nes07], they consider joint regularization by a strongly convex function which does not depend on the problem matrices $P, C$ but only on the constraint set $\mathcal{B}_{+,\infty}^n \times \Delta_p^+ \times \Delta_c^+$. Our area convex regularizer, on the other hand, is tailor made for the particular problem matrices $P, C$ as well as the constraint set.

# 4 Area Convexity for Mixed Packing Covering LPs

In this section, we present our technical results and algorithm for the MPC problem, with the end goal of proving Theorem 1.1. First, we relate an $(1 + \varepsilon)$-approximate solution to the saddle point problem to an $\varepsilon$-approximate solution to MPC. Next, we present some theoretical background towards the goal of choosing and analyzing an appropriate area-convex regularizer in the context of the saddle point formulation, where the key requirement of the area convex function is to obtain a provable and efficient convergence result. Finally, we explicitly show an area convex function which is generated using a simple "gadget" function. We show that this area convex function satisfies all key requirements and hence achieves the desired accelerated rate of convergence. This section closely follows [She17], in which the author chooses an area convex function specific to the undirected multicommodity flow problem. Due to space constraints, we relegate almost all proofs to Appendix A (in the full version) and simply include pointers to proofs in [She17] when it is directly applicable.

## 4.1 Saddle Point Formulation for MPC

Consider the saddle point formulation in (2) for MPC. Given a feasible primal-dual feasible solution pair $(x, y, z)$ and $(\bar{x}, \bar{y}, \bar{z})$ for (2), we denote $w = (x, u, y, z)$ and $\bar{w} = (\bar{x}, \bar{u}, \bar{y}, \bar{z})$ where $u, \bar{u} \in \mathbb{R}$. Then, we define a function $Q : \mathbb{R}^{n+1+p+c} \times \mathbb{R}^{n+1+p+c} \to \mathbb{R}$ as

$$Q(w, \bar{w}) \stackrel{\text{def}}{=} [\bar{y}^T\ \bar{z}^T] \begin{bmatrix} P & -\mathbf{1}_p \\ -C & \mathbf{1}_c \end{bmatrix} \begin{bmatrix} x \\ u \end{bmatrix} - [y^T\ z^T] \begin{bmatrix} P & -\mathbf{1}_p \\ -C & \mathbf{1}_c \end{bmatrix} \begin{bmatrix} \bar{x} \\ \bar{u} \end{bmatrix}.$$

Note that if $u = \bar{u} = 1$, then

$$\sup_{\bar{w} \in \mathcal{W}} Q(w, \bar{w}) = \sup_{\bar{x} \in \mathcal{B}^n_{+,\infty}, \bar{y} \in \Delta^+_p, \bar{z} \in \Delta^+_c} L(x, \bar{y}, \bar{z}) - L(\bar{x}, y, z)$$

is precisely the primal-dual gap function defined in Section 2. Notice that if $(x^*, y^*, z^*)$ is a saddle point of (2), then we have

$$L(x^*, y, z) \leqslant L(x^*, y^*, z^*) \leqslant L(x, y^*, z^*)$$

for all $x \in \mathcal{B}^n_{+,\infty}, y \in \Delta^+_p, z \in \Delta^+_c$. From above equation, it is clear that $Q(w, w^*) \geqslant 0$ for all $w \in \mathcal{W}$ where $\mathcal{W} \stackrel{\text{def}}{=} \mathcal{B}^n_{+,\infty} \times \{1\} \times \Delta^+_p \times \Delta^+_c$ and $w^* = (x^*, 1, y^*, z^*) \in \mathcal{W}$. Moreover, $Q(w^*, w^*) = 0$. This motivates the following accuracy measure of the candidate approximate solution $w$.

**Definition 4.1.** We say that $w \in \mathcal{W}$ is an $\varepsilon$-optimal solution of (2) iff

$$\sup_{\bar{w} \in \mathcal{W}} Q(w, \bar{w}) \leqslant \varepsilon.$$

**Remark 4.2.** *Recall the definition of $M_A$ for a matrix $A$ in Section 3. We can rewrite $Q(w, \bar{w}) = \bar{w}^T J w$ where $J = M_H$ and*

$$H = \begin{bmatrix} P & -\mathbf{1}_p \\ -C & \mathbf{1}_c \end{bmatrix} \quad \Rightarrow \quad J := \begin{bmatrix} \mathbf{0}_{n \times n} & \mathbf{0}_{n \times 1} & -P^T & C^T \\ \mathbf{0}_{1 \times n} & 0 & \mathbf{1}_p^T & -\mathbf{1}_c^T \\ P & -\mathbf{1}_p & \mathbf{0}_{p \times p} & \mathbf{0}_{p \times c} \\ -C & \mathbf{1}_c & \mathbf{0}_{c \times p} & \mathbf{0}_{c \times c} \end{bmatrix}.$$

*Thus, the gap function in Definition 4.1 can be written in the bilinear form $\sup_{\bar{w} \in \mathcal{W}} \bar{w}^T J w$.*

Lemma 4.3 relates the $\varepsilon$-optimal solution of (2) to the $\varepsilon$-approximate solution to MPC.

**Lemma 4.3.** *Let $(x, y, z)$ satisfy $\sup_{(\bar{x}, \bar{y}, \bar{z}) \in \mathcal{B}^n_{+,\infty} \times \Delta^+_p \times \Delta^+_c} L(x, \bar{y}, \bar{z}) - L(\bar{x}, y, z) \leqslant \varepsilon$. Then either*

*1. $x$ is an $\varepsilon$-approximate solution of MPC, or*
*2. $y, z$ satisfy $y^T(P\bar{x} - \mathbf{1}_p) + z^T(-C\bar{x} + \mathbf{1}_c) > 0$ for all $\bar{x} \in \mathcal{B}^n_{+,\infty}$.*

This lemma states that in order to find an $\varepsilon$-approximate solution of MPC, it suffices to find $\varepsilon$-optimal solution of (2). Henceforth, we will focus on $\varepsilon$-optimality of the saddle point formulation (2).

## 4.2 Area Convexity with Saddle Point Framework

Here we state some useful lemmas which help in determining whether a differentiable function is area convex. We start with the following remark which follows from the definition of area convexity (Definition 3.2).

**Remark 4.4.** *If $\phi$ is area convex with respect to $A$ on a convex set $K$, and $\bar{K} \subseteq K$ is a convex set, then $\phi$ is area convex with respect to $A$ on $\bar{K}$.*

The following two lemmas from [She17] provide the key characterization of area convexity.

**Lemma 4.5.** *Let $A \in \mathbb{R}^{2\times2}$ symmetric matrix. $A \succeq_{\mathrm{i}} \begin{bmatrix} 0 & -1 \\ 1 & 0 \end{bmatrix} \Leftrightarrow A \succeq 0$ and $\det(A) \geqslant 1$.*

**Lemma 4.6.** *Let $\phi$ be twice differentiable on the interior of convex set $K$, i.e., $\mathrm{int}(K)$.*

1. *If $\phi$ is area convex with respect to $A$ on $\mathrm{int}(K)$, then $d^2\phi(x) \succeq_{\mathrm{i}} M_A$ for all $x \in \mathrm{int}(K)$.*

2. *If $d^2\phi(x) \succeq_{\mathrm{i}} M_A$ for all $x \in \mathrm{int}(K)$, then $\phi$ is area convex with respect to $\frac{1}{3}A$ on $\mathrm{int}(K)$. Moreover, if $\phi$ is continuous on $\mathrm{cl}(K)$, then $\phi$ is area convex with respect to $\frac{1}{3}A$ on $\mathrm{cl}(K)$.*

In order to handle the operator $\succeq_{\mathrm{i}}$ (recall from Section 2), we state some basic but important properties of this operator, which will come in handy in later proofs.

**Remark 4.7.** *For symmetric matrices $A$ and $C$ and antisymmetric matrices $B$ and $D$,*

1. *If $A \succeq_{\mathrm{i}} B$ then $A \succeq_{\mathrm{i}} (-B)$.*

2. *If $A \succeq_{\mathrm{i}} B$ and $\lambda \geqslant 0$ then $\lambda A \succeq_{\mathrm{i}} \lambda B$.*

3. *If $A \succeq_{\mathrm{i}} B$ and $C \succeq_{\mathrm{i}} D$ then $A + C \succeq_{\mathrm{i}} (B + D)$.*

Having laid a basic foundation for area convexity, we now focus on its relevance to solving the saddle point problem (2). Considering Remark 4.2, we can write the gap function criterion of optimality in terms of bilinear form of the matrix $J$. Suppose we have a function $\phi$ which is area convex with respect to $H$ on set $\mathcal{W}$. Then, consider the following *jointly-regularized* version of the bilinear form:

$$\tilde{\eta}(w) := \sup_{\bar{w} \in \mathcal{W}} \bar{w}^T J w - \phi(\bar{w}). \tag{3}$$

Similar to Nesterov's dual extrapolation, one can attain $O(1/\varepsilon)$ convergence of accelerated gradient descent for function $\tilde{\eta}(w)$ in (3) over variable $w$. In order to obtain gradients of $\tilde{\eta}(w)$, we need access to $\mathrm{argmax}_{\bar{w} \in \mathcal{W}} \bar{w}^T J w - \phi(\bar{w})$. However, it may not be possible to find an exact maximizer in all cases. Again, one can get around this difficulty by instead using an approximate optimization oracle of the problem in (3).

**Definition 4.8.** A $\delta$-optimal solution oracle (OSO) for $\phi : \mathcal{W} \to \mathbb{R}$ takes input $a$ and outputs $w \in \mathcal{W}$ such that

$$a^T w - \phi(w) \geqslant \sup_{\bar{w} \in \mathcal{W}} a^T \bar{w} - \phi(\bar{w}) - \delta.$$

Given $\Phi$ as a $\delta$-OSO for a function $\phi$, consider the following algorithm (Algorithm 4.2):

---
**Algorithm 1** Area Convex Mixed Packing Covering (AC-MPC)

---
Initialize $w_0 = (\mathbf{0}_n, 1, \mathbf{0}_{p+c})$
**for** $t = 0, \ldots, T$ **do**
    $w_{t+1} \leftarrow w_t + \Phi(Jw_t + 2J\Phi(Jw_t))$
**end for**

---

For Algorithm 4.2, [She17] shows the following:

**Lemma 4.9.** *Let $\phi : \mathcal{W} \to [-\rho, 0]$. Suppose $\phi$ is area convex with respect to $2\sqrt{3}H$ on $\mathcal{W}$. Then for $J = M_H$ and for all $t \geqslant 1$ we have $w_t/t \in \mathcal{W}$ and,*

$$\sup_{\bar{w} \in \mathcal{W}} \bar{w} J \frac{w_t}{t} \leqslant \delta + \frac{\rho}{t}.$$

*In particular, in $\frac{\rho}{\varepsilon}$ iterations, Algorithm 4.2 obtain $(\delta + \varepsilon)$-solution of the saddle point problem (2).*

The analysis of this lemma closely follows the analysis of Nesterov's dual extrapolation.

Note that, each iteration consists of $O(1)$ matrix-vector multiplications, $O(1)$ vector additions, and $O(1)$ calls to the approximate oracle. Since the former two are parallelizable to $O(\log n)$ depth, the same remains to be shown for the oracle computation to complete the proof of the run-time in Theorem 1.1.

Recall from the discussion in Section 3 that the critical bottleneck of Nesterov's method is that diameter of the $\ell_\infty$ ball is $\Omega(\sqrt{n})$, which is achieved even in the Euclidean $\ell_2$ norm. This makes $\rho$ in Lemma 4.9 to also be $\Omega(\sqrt{n})$, which can be a major bottleneck for high dimensional LPs, which are commonplace among real-world applications.

Although, on the face of it, area convexity applied to the saddle point formulation (2) has a similar framework to Nesterov's dual extrapolation, the challenge is to construct a $\phi$ for which we can overcome the above bottleneck. Particularly, there are three key challenges to tackle:
**1.** We need to show that existence of a function $\phi$ that is area convex with respect to $H$ on $\mathcal{W}$.
**2.** $\phi : \mathcal{W} \to [-\rho, 0]$ should be such that $\rho$ is not too large.
**3.** There should exist an efficient $\delta$-OSO for $\phi$.
In the next subsection, we focus on these three aspects in order to complete our analysis.

### 4.3   Choosing an area convex function

First, we consider a simple 2-D gadget function and prove a "nice" property of this gadget. Using this gadget, we construct a function which can be shown to be area convex using the aforementioned property of the gadget.

Let $\gamma_\beta : \mathbb{R}_+^2 \to \mathbb{R}$ be a function parameterized by $\beta$ defined as

$$\gamma_\beta(a, b) = ba \log a + \beta b \log b.$$

**Lemma 4.10.** *Suppose* $\beta \geqslant 2$. *Then* $d^2 \gamma_\beta(a, b) \succeq \begin{bmatrix} 0 & -1 \\ 1 & 0 \end{bmatrix}$ *for all* $a \in (0, 1]$ *and* $b > 0$.

Now, using the function $\gamma_\beta$, we construct a function $\phi$ and use the sufficiency criterion provided in Lemma 4.6 to show that $\phi$ is area convex with respect to $J$ on $\mathcal{W}$. Note that our set of interest $\mathcal{W}$ is not full-dimensional, whereas Lemma (4.6) is only stated for int and not for relint. To get around this difficulty, we consider a larger set $\overline{\mathcal{W}} \supset \mathcal{W}$ such that $\overline{\mathcal{W}}$ is full dimensional and $\phi$ is area convex on $\overline{\mathcal{W}}$. Then we use Remark 4.4 to obtain the final result, i.e., area convexity of $\phi$.

**Theorem 4.11.** *Let* $w = (x, u, y, z)$ *and define*
$$\phi(w) \overset{\text{def}}{=} \sum_{i=1}^{p} \sum_{j=1}^{n} P_{ij} \gamma_{p_i}(x_j, y_i) + \sum_{i=1}^{p} \gamma_2(u, y_i) + \sum_{i=1}^{c} \sum_{j=1}^{n} C_{ij} \gamma_{c_i}(x_j, z_i) + \sum_{i=1}^{c} \gamma_2(u, z_i),$$
*where* $p_i = 2 * \frac{\|P\|_\infty}{\|P_{i,:}\|_1}$ *and* $c_i = 2 * \frac{\|C\|_\infty}{\|C_{i,:}\|_1}$, *then* $\phi$ *is area convex with respect to* $\frac{1}{3} \begin{bmatrix} P & -\mathbf{1}_p \\ -C & \mathbf{1}_c \end{bmatrix}$ *on set* $\overline{\mathcal{W}} := \mathcal{B}_{+,\infty}^{n+1}(1) \times \Delta_p^+ \times \Delta_c^+$. *In particular, it also implies* $6\sqrt{3}\phi$ *is area convex with respect to* $2\sqrt{3} \begin{bmatrix} P & -\mathbf{1}_p \\ -C & \mathbf{1}_c \end{bmatrix}$ *on set* $\mathcal{W}$.

Theorem 4.11 addresses the first part of the key three challenges. Next, Lemma 4.12 shows an upper bound on the range of $\phi$.

**Lemma 4.12.** *Function* $\phi : \mathcal{W} \to [-\rho, 0]$ *then* $\rho = O(\|P\|_\infty \log p + \|C\|_\infty \log c)$.

Finally, we need an efficient $\delta$-OSO. Consider the following alternating minimization algorithm.

---

**Algorithm 2** $\delta$-OSO for $\phi$

---

Input $a \in \mathbb{R}^{n+1}, a^1 \in \mathbb{R}^p, a^2 \in \mathbb{R}^c, \delta > 0$
Initialize $(x^0, u^0) \in \mathcal{B}_{+,\infty}^n \times \{1\}$ arbitrarily.
**for** $k = 1, \ldots, K$ **do**
    $(y^k, z^k) \leftarrow \underset{y \in \Delta_c^+, \ z \in \Delta_p^+}{\text{argmax}} \ y^T a^1 + z^T a^2 - \phi(x^{k-1}, u^{k-1}, y, z)$
    $(x^k, u^k) \leftarrow \underset{(x,u) \in \mathcal{B}_{+,\infty}^n \times \{1\}}{\text{argmax}} \ [x^T \ u]a - \phi(x, u, y^k, z^k)$
**end for**

---

[Bec15] shows the following convergence result.

**Lemma 4.13.** *For $\delta > 0$, Algorithm 2 is a $\delta$-OSO for $\phi$ which converges in $O(\log \frac{1}{\delta})$ iterations.*

We show that for our chosen $\phi$, we can perform the two argmax computations in each iteration of Algorithm 2 analytically in time $O(\text{nnz}(P) + \text{nnz}(C))$, and hence we obtain a $\delta$-OSO which takes $O((\text{nnz}(P) + \text{nnz}(C) \log \frac{1}{\delta})$ total work. Parallelizing matrix-vector multiplications eliminates the dependence on $\text{nnz}(P)$ and $\text{nnz}(C)$, at the cost of another $\log(N)$ term.

**Lemma 4.14.** *Each* argmax *in Algorithm 2 can be computed as follows:*
$x^k = \min\{\exp(\frac{a}{P^T y^k + C^T z^k} - 1), \mathbf{1}_n\}$ *for all* $j \in [n]$.
$y^k = \text{proj}_{\Delta_p^+}\big(\exp\{\frac{1}{2(\|P\|_\infty + 1)}(a^1 - Px^{k-1} \log x^{k-1})\}\big)$
$z^k = \text{proj}_{\Delta_c^+}\big(\exp\{\frac{1}{2(\|C\|_\infty + 1)}(a^2 - Cx^{k-1} \log x^{k-1})\}\big)$
*In particular, we can compute* $x^k, y^k, z^k$ *in* $O(\text{nnz}(P) + \text{nnz}(C))$ *work and* $O(\log N)$ *parallel time.*

## Acknowledgements

We thank Richard Peng for many important pointers and discussions.

## Footnotes

*Work done when author was at Georgia Tech.

[2]$d$ here is the maximum number of constraints that any variable appears in.

[3]This transformation can be achieved by adapting techniques from [WRM16] while increasing dimension of the problem up to a logarithmic factor. Details of this fact are in Appendix B in the full version of this paper. For the purpose of the main text, we work with this assumption.

[4]Definitions of Lipschitz-smoothness and strong convexity can be found in many texts in nonlinear programming and machine learning. e.g. [Bub14]. Intuitively, $f$ is Lipschitz-smooth if the rate of change of $\nabla f$ can be bounded by a quantity known as the "constant of Lipschitz smoothness".

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
