[Supplementary Material]

# Faster width-dependent algorithm for mixed packing and covering LPs

Digvijay Boob [1]  Saurabh Sawlani[2]   Di Wang[3]

[1]Inudstrial and Systems Engineering, Gerogia Tech
[2]College of Computing, Georgia Tech
[3]Google Research

Oral Presentation at NeurIPS 2019
December, 2019

# Outline

1. Problem of interest

2. Technical Overview

3. Area Convexity

4. Summary

## Mixed Packing and Covering(MPC) LP

Does there exists an $x \in \square^n := \{x \geqslant \mathbf{0}_n, \|x\|_\infty \leqslant 1\}$ such that

$$Px \leqslant \mathbf{1}_p, \qquad \text{(Packing constraints)},$$
$$Cx \geqslant \mathbf{1}_c, \qquad \text{(Covering Constraints)},$$

where $P, C \geqslant 0$.

**Def:** We say that $x$ is an $\varepsilon$-approximate solution to the MPC problem if $x$ satisfies $x \in \square^n, Px \leqslant (1+\varepsilon)\mathbf{1}_p, Cx \geqslant (1-\varepsilon)\mathbf{1}_c$.

## Mixed Packing and Covering(MPC) LP

Does there exists an $x \in \square^n := \{x \geqslant \mathbf{0}_n, \|x\|_\infty \leqslant 1\}$ such that

$$Px \leqslant \mathbf{1}_p, \qquad \text{(Packing constraints)},$$
$$Cx \geqslant \mathbf{1}_c, \qquad \text{(Covering Constraints)},$$

where $P, C \geqslant 0$.

**Def:** We say that $x$ is an $\varepsilon$-approximate solution to the MPC problem if $x$ satisfies $x \in \square^n, Px \leqslant (1+\varepsilon)\mathbf{1}_p, Cx \geqslant (1-\varepsilon)\mathbf{1}_c$.

## Application: Optimal Transport Problem

Optimal transport is a problem of computing Wasserstein distance between two n-dimensional distributions.

## Application: Optimal Transport Problem

Optimal transport is a problem of computing Wasserstein distance between two n-dimensional distributions.

Modeled as LP:

$$\min_{X} \langle C, X \rangle$$
$$\text{s.t. } \{X \geqslant 0, X\mathbf{1} = u, X^{T}\mathbf{1} = v\}.$$

## Motivation

Mixed Packing-Covering LPs

| Pure Packing | | Zero-sum Matrix Games |
|---|---|---|
| Bipartite matching | | Optimal Transport |
| | | Multi-commodity flow |
| Pure Covering | | Mechanism Design |
| | | Positive Linear Systems |
| Minimum Set Cover | | Scheduling |
| | | X-Ray Tomography |

## Previous Results

**Def:** Width $w$ is maximum non-zeros in any row of $P$ or $C$.

Table: Runtime for obtaining $\varepsilon$-approximate solution:

|  | Runtime | Comments |
|---|---|---|
| Nesterov | $\tilde{O}(w\sqrt{n}\varepsilon^{-1})$ | width-dependent |
| Young 2014 | $\tilde{O}(\varepsilon^{-4})$ | |
| Mahoney *et al.* | $\tilde{O}(\varepsilon^{-3})$ | |
| **Our work** | $\tilde{O}(w\varepsilon^{-1})$ | width-dependent |

## Previous Results

**Def:** Width $w$ is maximum non-zeros in any row of $P$ or $C$.

Table: Runtime for obtaining $\varepsilon$-approximate solution:

|                | Runtime                                  | Comments        |
|----------------|------------------------------------------|-----------------|
| Nesterov       | $\widetilde{O}(w\sqrt{n}\varepsilon^{-1})$ | width-dependent |
| Young 2014     | $\widetilde{O}(\varepsilon^{-4})$         |                 |
| Mahoney *et al.* | $\widetilde{O}(\varepsilon^{-3})$       |                 |
| **Our work**   | $\widetilde{O}(w\varepsilon^{-1})$        | width-dependent |

# Saddle Point Problem (SPP) Reformulation

- Reformulate the MPC problem as a SPP:
$$\min_{x \in \square^n} \max_{y \in \Delta^p, z \in \Delta^c} L(x, y, z)$$

- $u := (x, y, z)$ and $\mathcal{U} := \square^n \times \Delta^p \times \Delta^c$.

- Convergence: $u \in \mathcal{U}$ s.t. **primal-dual gap** $\text{Gap}(u) := \sup_{\bar{u} \in \mathcal{U}} L(x, \bar{y}, \bar{z}) - L(\bar{x}, y, z)$ is small ( $Q(u) \leqslant \varepsilon$ ).

- $u$ is $\varepsilon$-SPP then either
  1. $x$ is an $\varepsilon$-approx solution to MPC, or
  2. We obtain a certificate of infeasibility.

$\Delta^3$

(0,0,1)

(0,1,0)

(1,0,0)

$\square^3$

(0, 1, 1)

(0, 0, 1)          (1, 1, 1)

(1, 0, 1)          (0, 1, 0)

(0, 0, 0)          (1, 1, 0)

(1, 0, 0)

# Saddle Point Problem (SPP) Reformulation

- Reformulate the MPC problem as a SPP:
$$\min_{x \in \square^n} \max_{y \in \Delta^p, z \in \Delta^c} L(x, y, z)$$

- $u := (x, y, z)$ and $\mathcal{U} := \square^n \times \Delta^p \times \Delta^c$.

- Convergence: $u \in \mathcal{U}$ s.t. **primal-dual gap**
  $\text{Gap}(u) := \sup_{\bar{u} \in \mathcal{U}} L(x, \bar{y}, \bar{z}) - L(\bar{x}, y, z)$ is
  small ( $Q(u) \leqslant \varepsilon$ ).

- $u$ is $\varepsilon$-SPP then either
  1. $x$ is an $\varepsilon$-approx solution to MPC, or
  2. We obtain a certificate of infeasibility.

$\Delta^3$

(0,0,1)

(1,0,0)

(0,1,0)

$\square^3$

(0, 1, 1)

(0, 0, 1)          (1, 1, 1)

(1, 0, 1)

(0, 1, 0)

(0, 0, 0)          (1, 1, 0)

(1, 0, 0)

# Saddle Point Problem (SPP) Reformulation

- Reformulate the MPC problem as a SPP:
$$\min_{x \in \square^n} \max_{y \in \Delta^p, z \in \Delta^c} L(x, y, z)$$

- $u := (x, y, z)$ and $\mathcal{U} := \square^n \times \Delta^p \times \Delta^c$.

- Convergence: $u \in \mathcal{U}$ s.t. **primal-dual gap**
$\text{Gap}(u) := \sup_{\bar{u} \in \mathcal{U}} L(x, \bar{y}, \bar{z}) - L(\bar{x}, y, z)$ is
small ( $Q(u) \leqslant \varepsilon$).

- $u$ is $\varepsilon$-SPP then either
  1. $x$ is an $\varepsilon$-approx solution to MPC, or
  2. We obtain a certificate of infeasibility.

$\Delta^3$

$(0,0,1)$

$(1,0,0)$

$(0,1,0)$

$\square^3$

$(0, 1, 1)$

$(0, 0, 1)$        $(1, 1, 1)$

$(1, 0, 1)$

$(0, 1, 0)$

$(0, 0, 0)$

$(1, 0, 0)$        $(1, 1, 0)$

## Saddle Point Problem (SPP) Reformulation

$\Delta^3$

(0,0,1)

- Reformulate the MPC problem as a SPP:
$$\min_{x \in \square^n} \max_{y \in \Delta^p, z \in \Delta^c} L(x, y, z)$$

- $u := (x, y, z)$ and $\mathcal{U} := \square^n \times \Delta^p \times \Delta^c$.

(1,0,0)

- Convergence: $u \in \mathcal{U}$ s.t. **primal-dual gap**
$\text{Gap}(u) := \sup_{\bar{u} \in \mathcal{U}} L(x, \bar{y}, \bar{z}) - L(\bar{x}, y, z)$ is
small ( $Q(u) \leqslant \varepsilon$).

(0,1,0)

- $u$ is $\varepsilon$-SPP then either
  1. $x$ is an $\varepsilon$-approx solution to MPC, or
  2. We obtain a certificate of infeasibility.

$\square^3$

## Standard Methods

- General Problem: $\min_{w \in X} f(w)$

- Regularized problem: $\min_{w \in X} f(w) + \phi(w)$

- $\phi$ is strongly convex on $X$.

- Rate of convergence: e.g. Nesterov's accelerated methods:
  $$\frac{1}{\varepsilon} \times \text{range of } \phi \text{ on set } X.$$

# Standard Methods

- General Problem: $\min_{w \in X} f(w)$
- Regularized problem: $\min_{w \in X} f(w) + \phi(w)$
- $\phi$ is strongly convex on $X$.
- Rate of convergence: e.g. Nesterov's accelerated methods:
  $\frac{1}{\varepsilon} \times$ range of $\phi$ on set $X$.

# Standard Methods

- General Problem: $\min_{w \in X} f(w)$
- Regularized problem: $\min_{w \in X} f(w) + \phi(w)$
- $\phi$ is strongly convex on $X$.
- Rate of convergence: e.g. Nesterov's accelerated methods:
$$\frac{1}{\varepsilon} \times \text{range of } \phi \text{ on set } X.$$

## Standard Methods + MPC

- General Problem:     $\min\limits_{w \in X} f(w) + \phi(w)$

- For the case of MPC problems:
    1. $f$ is primal-dual gap $Q$.
    2. $w$ is joint primal-dual variable $u$.
    3. $X$ is joint domain $\mathcal{U}(= \square^n \times \Delta^p \times \Delta^c)$.
    4. $\phi$ is strongly convex regularizers $\phi_1(x) + \phi_2(y) + \phi_3(z)$.

- Algorithm of choice: Nesterov's Dual Extrapolation

- Range of regularizers $\tilde{\Theta}(w\sqrt{n})$

- Range above tight for strongly convex regularizer on $\square^n$.

# Standard Methods + MPC

- General Problem: $\min\limits_{w \in X} f(w) + \phi(w)$

- For the case of MPC problems:

  1. $f$ is primal-dual gap $Q$.
  2. $w$ is joint primal-dual variable $u$.
  3. $X$ is joint domain $\mathcal{U}(= \square^n \times \Delta^p \times \Delta^c)$.
  4. $\phi$ is strongly convex regularizers $\phi_1(x) + \phi_2(y) + \phi_3(z)$.

- Algorithm of choice: Nesterov's Dual Extrapolation

- Range of regularizers $\tilde{\Theta}(w\sqrt{n})$

- Range above tight for strongly convex regularizer on $\square^n$.

# Standard Methods + MPC

- General Problem: $\quad \min\limits_{w \in X} f(w) + \phi(w)$

- For the case of MPC problems:

  1. $f$ is primal-dual gap $Q$.
  2. $w$ is joint primal-dual variable $u$.
  3. $X$ is joint domain $\mathcal{U}(= \Box^n \times \Delta^p \times \Delta^c)$.
  4. $\phi$ is strongly convex regularizers $\phi_1(x) + \phi_2(y) + \phi_3(z)$.

- Algorithm of choice: Nesterov's Dual Extrapolation

- Range of regularizers $\widetilde{\Theta}(w\sqrt{n})$

- Range above tight for strongly convex regularizer on $\Box^n$.

# Standard Methods + MPC

- General Problem: $\min\limits_{w \in X} f(w) + \phi(w)$

- For the case of MPC problems:
  1. $f$ is primal-dual gap $Q$.
  2. $w$ is joint primal-dual variable $u$.
  3. $X$ is joint domain $\mathcal{U}(= \square^n \times \Delta^p \times \Delta^c)$.
  4. $\phi$ is strongly convex regularizers $\phi_1(x) + \phi_2(y) + \phi_3(z)$.

- Algorithm of choice: Nesterov's Dual Extrapolation

- Range of regularizers $\widetilde{\Theta}(w\sqrt{n})$

- Range above tight for strongly convex regularizer on $\square^n$.

## Standard Methods + MPC

- General Problem:  $\min\limits_{w \in X} f(w) + \phi(w)$

- For the case of MPC problems:
    1. $f$ is primal-dual gap $Q$.
    2. $w$ is joint primal-dual variable $u$.
    3. $X$ is joint domain $\mathcal{U}(= \square^n \times \Delta^p \times \Delta^c)$.
    4. $\phi$ is strongly convex regularizers $\phi_1(x) + \phi_2(y) + \phi_3(z)$.

- Algorithm of choice: Nesterov's Dual Extrapolation

- Range of regularizers $\underbrace{\widetilde{\Theta}(w\sqrt{n})}_{\text{Tight}}$

- Range above tight for strongly convex regularizer on $\square^n$ .

## Area Convexity

Area convexity [Sherman 2017] :

1. Is weaker than strong convexity. One can obtain area convex regularizer with small range over $\ell_\infty$-ball.

2. Still good enough to obtain $O(\text{range of regularizer} \times \frac{1}{\varepsilon})$ convergence.

## Definition

- Strong convexity: for all $t, u \in K$
  $\phi(\frac{t+u}{2}) \leqslant \frac{1}{2}(\phi(t) + \phi(u)) - \frac{1}{2}\|t - u\|^2$.

- **Def:** A function $\phi$ is area convex w.r.t. matrix $M$ on convex set $K$ iff for any $t, u, v \in K$,
  $\phi(\frac{t+u+v}{3}) \leqslant \frac{1}{3}\big(\phi(t) + \phi(u) + \phi(v)\big) - \frac{1}{3\sqrt{3}}\underbrace{(v - u)^T M(u - t)}_{\text{'area' of } \Delta(tuv)}$.

## Definition

- Strong convexity: for all $t, u \in K$
  $\phi(\frac{t+u}{2}) \leqslant \frac{1}{2}(\phi(t) + \phi(u)) - \frac{1}{2}\|t - u\|^2$.

- **Def:** A function $\phi$ is area convex w.r.t. matrix $M$ on convex set $K$ iff for any $t, u, v \in K$,
  $\phi(\frac{t+u+v}{3}) \leqslant \frac{1}{3}\big(\phi(t) + \phi(u) + \phi(v)\big) - \frac{1}{3\sqrt{3}} \underbrace{(v - u)^T M(u - t)}_{\text{'area' of } \Delta(tuv)}$.

## An Example

- For any $t, u \in K$ , area convex $\phi$ requires mere convexity: $\phi(\frac{t+u}{2}) \leqslant \frac{1}{2}(\phi(t) + \phi(u))$.
- Consider $\gamma(x, y) = yx \log x + 2y \log y$.

  Area convex w.r.t. $\begin{bmatrix} 0 & -1 \\ 1 & 0 \end{bmatrix}$ on set $0 \leqslant x, y \leqslant 1$.

Figure: Auxiliary view

Figure: Level set $\gamma(x, y) \leqslant -0.5$

## Area Convexity + MPC

- Use $\phi : \mathcal{U} \rightarrow [-\rho, 0]$ as area convex regularizer w.r.t. a matrix depending on $P$ and $C$ on set $\mathcal{U}$.

- Area convexity: relaxed requirement, we can show $\phi$ for which $\rho = O(\|P\|_\infty \log p + \|C\|_\infty \log c)$

- This $\rho$ is of order width, $w$ of MPC, **gets rid of the $\sqrt{n}$ factor**.

# Area Convexity + MPC

- Use $\phi : \mathcal{U} \to [-\rho, 0]$ as area convex regularizer w.r.t. a matrix depending on $P$ and $C$ on set $\mathcal{U}$.
- Area convexity: relaxed requirement, we can show $\phi$ for which $\rho = O(\|P\|_\infty \log p + \|C\|_\infty \log c)$
- This $\rho$ is of order width, $w$ of MPC, **gets rid of the $\sqrt{n}$ factor**.

# Area Convexity + MPC

- Use $\phi : \mathcal{U} \to [-\rho, 0]$ as area convex regularizer w.r.t. a matrix depending on $P$ and $C$ on set $\mathcal{U}$.
- Area convexity: relaxed requirement, we can show $\phi$ for which $\rho = O(\|P\|_\infty \log p + \|C\|_\infty \log c)$
- This $\rho$ is of order width, $w$ of MPC, **gets rid of the $\sqrt{n}$ factor**.

## Salient Features of Our Regularizer

- Standard regularization: $\phi_1(x) + \phi_2(y)$.
- Our regularization contains terms of the following type:

$$y_j P_{ij} x_j \log x_j.$$

- Interaction of dual variable $y$ and primal variable $x$.
- Standard case: separate regularization of primal and dual variable as $\phi_1(x)$ and $\phi_2(y)$.
- Depends on the problem matrix $P$ and $C$. Explores the structure of the problem
- Independent to problem matrix $P$ and $C$.

## Salient Features of Our Regularizer

- Standard regularization: $\phi_1(x) + \phi_2(y)$.

- Our regularization contains terms of the following type:

$$y_j P_{ij} x_j \log x_j.$$

- Interaction of dual variable $y$ and primal variable $x$.

- Standard case: separate regularization of primal and dual variable as $\phi_1(x)$ and $\phi_2(y)$.

- Depends on the problem matrix $P$ and $C$. Explores the structure of the problem

- Independent to problem matrix $P$ and $C$.

## Salient Features of Our Regularizer

- Standard regularization: $\phi_1(x) + \phi_2(y)$.
- Our regularization contains terms of the following type:

$$y_j P_{ij} x_j \log x_j.$$

- Interaction of dual variable $y$ and primal variable $x$.
- Standard case: separate regularization of primal and dual variable as $\phi_1(x)$ and $\phi_2(y)$.
- Depends on the problem matrix $P$ and $C$. Explores the structure of the problem
- Independent to problem matrix $P$ and $C$.

# Summary

- Strongly convex regularizer and $\ell_\infty$-barrier.

- Explicit area convex regularizer for MPC which circumvents the $\ell_\infty$-barrier.

  1. Area convexity weaker than strong convexity. Range of the regularizer can be made $\tilde{O}(w)$ on $\ell_\infty$-ball

  2. Still suffices to obtain $\tilde{O}(\frac{w}{\varepsilon})$ convergence.

- See more details:

  Poster: #232

  Paper: *Faster width-dependent algorithms for mixed packing and covering LPs, arXiv 2019.*

## Summary

- Strongly convex regularizer and $\ell_\infty$-barrier.
- Explicit area convex regularizer for MPC which circumvents the $\ell_\infty$-barrier.
  1. Area convexity weaker than strong convexity. Range of the regularizer can be made $\tilde{O}(w)$ on $\ell_\infty$-ball
  2. Still suffices to obtain $\tilde{O}(\frac{w}{\varepsilon})$ convergence.
- See more details:

  Poster: #232

  Paper: *Faster width-dependent algorithms for mixed packing and covering LPs, arXiv 2019.*

## Summary

- Strongly convex regularizer and $\ell_\infty$-barrier.
- Explicit area convex regularizer for MPC which circumvents the $\ell_\infty$-barrier.
    1. Area convexity weaker than strong convexity. Range of the regularizer can be made $\tilde{O}(w)$ on $\ell_\infty$-ball
    2. Still suffices to obtain $\tilde{O}(\frac{w}{\varepsilon})$ convergence.
- See more details:

    Poster: #**232**

    Paper: *Faster width-dependent algorithms for mixed packing and covering LPs, arXiv 2019.*

Thanks!

# Questions?