[Reviews · NeurIPS 2019]

Reviewer 1



1. Page 4, the ell_{infty} barrier. The limitations of performing first-order methods over the ell_{infty} ball go beyond the fact that strongly convex functions grow too fast (Lemma 3.1): there are provable oracle complexity lower bounds for this problem (see Guzman & Nemirovski, "On Lower Complexity Bounds for Large-Scale Smooth Convex Optimization" 2015). I believe Lemma 3.1 is great to make a point, but it is also interesting to point out the provable lower bounds that area convexity circumvents. 1. Page 6, line 215. "is precisely the primal-dual gap". I don't think this is the case, since the primal-dual gap should be the supremum over \bar{w} variable, not just for any primal-dual pair. Could the authors clarify this? 2. Page 6, definition 4.1. In the same spirit as the previous comment: What is the relation between an eps-optimal solution and the duality gap? 3. Page 7, eqn (3). Shouldn't the supremum be over \bar{w}, instead of \bar{x}? 4. Page 7, line 253. I believe \bar{\eta} is exactly the function defined in (3). If this is the case, it would be better to define it in the same equation 5. Page 7, algorithm 1. It is mentioned that the algorithm is related to dual extrapolation, but it also looks similar to the extragradient method (a.k.a. Mirror-Prox). Are there connections in this direction? 6. Page 11, line 381. This is a very minor detail, but after assuming there exists a feasible \tilde{x}, shouldn't the second case start with: Suppose no \tilde{x} is feasible? (as opposed to just x). Starting from x is formally correct, but it is a bit confusing, given the assumption of the previous paragraph. 7. Page 11, proof of Lemma 4.5. This proof needs some polishing. I believe one can directly appeal to Schur complements, and the first two lines are just confusing. Besides, there is a typo on the inverse of A: a_{22}=a, and not d.

Reviewer 2



The paper presents an original application of the novel framework of area-convexity to the design of accelerated algorithms for mixed packing and covering LPs that break the \ell_\infty regularization barrier. It is very significant in that it extends the area-convexity idea beyond the original applications in Sherman's 2017 paper. I believe it will lead to more scholars getting to know this line of work and deploying it for their problems. The paper is well-written.

Reviewer 3



Originality: Maybe the idea is applying [She17] to MLP itself is not very hard to come up with, doing it right seems to be non-trivial. Quality: I didn't find any flaw in the proof as far as I checked. Clarity: The paper is well written and easy to follow. Significance: As I mentioned, it seems the obtained result is significant as the dependency on eps is often problematic.

[Author Response · NeurIPS 2019]

Dear reviewers, we greatly appreciate your remarks and suggestions. We will address the comments in the following.

1. **Page 4, the $\ell_\infty$ barrier. The limitations of performing first-order methods over the $\ell_\infty$ ball go beyond the fact that strongly convex functions grow too fast (Lemma 3.1): there are provable oracle complexity lower bounds for this problem (see Guzman and Nemirovski, "On Lower Complexity Bounds for Large-Scale Smooth Convex Optimization" 2015).**
   Thank you for pointing out the connection to the result by Guzman and Nemirovski, and their lower bound result for smooth convex minimization over $\ell_\infty$-ball is indeed very interesting in this context. We will include a brief discussion on this point after Lemma 3.1 in the updated version.

2. **Page 6, line 215. "is precisely the primal-dual gap". I don't think this is the case, since the primal-dual gap should be the supremum over $\overline{w}$ variable. Could the authors clarify this?**
   Thanks for spotting this typo. We indeed meant that taking a $\sup$ over the $\overline{w}$ variable gives the primal-dual gap of the solution $(x, y, z)$ as defined earlier in line 109. We will correct this accordingly.

3. **Page 6, definition 4.1. In the same spirit as the previous comment: What is the relation between an eps-optimal solution and the duality gap?**
   If we have an $\epsilon$-optimal solution of Eq(2) (i.e., Definition 4.1), we can read from it a solution $(x, y, z)$ whose duality gap is at most $\epsilon$ (as defined in Line 109). We further show (in Lemma 4.3) that such a solution translates to either a $\epsilon$-approximate solution $x$ to the original mixed packing-covering LP, or a certificate (with $(y, z)$) that the original MPC is infeasible.

4. **Page 7, eqn (3). Shouldn't the supremum be over $\overline{w}$, instead of $\overline{x}$? Page 7, line 253. I believe $\widetilde{\eta}$ is exactly the function defined in (3).**
   Eq(3): Yes, the supremum should be over $\overline{w} \in \mathcal{W}$ instead of $\overline{x}$, and $\widetilde{\eta}$ is the same function defined in (3). We will correct the typo, and clarify the definition in the revised version.

5. **Page 7, algorithm 1. It is mentioned that the algorithm is related to dual extrapolation, but it also looks similar to the extra-gradient method (a.k.a. Mirror-Prox). Are there connections in this direction?**
   As discussed in the dual-extrapolation paper, Nesterov's method was on a high level motivated by Mirror-Prox. The two methods can be the same if the setup is Euclidean and the underlying space is unconstrained, but in general cases they give very different algorithms. Essentially dual-extrapolation carries out extra-gradient steps in the dual space. In that spirit, the connection is similar to the case of gradient descent vs mirror decent.

6. **Page 11, line 381. This is a very minor detail, but after assuming there exists a feasible $\widetilde{x}$, shouldn't the second case start with: Suppose no $\widetilde{x}$ is feasible? (as opposed to just x). Starting from x is formally correct, but it is a bit confusing, given the assumption of the previous paragraph**
   We agree the wording is a bit confusing, and indeed the first half is mostly redundant. The succinct argument should be that if the $x$ we obtained is an $\epsilon$-approximate solution to the MPC instance, then we are done, i.e. case (1). In the second case, i.e. if our $x$ is not an $\varepsilon$-approximate solution, we certify that the original MPC instance is infeasible. The first half is proving the additional statement that if MPC is feasible then $x$ must be $\varepsilon$-approximate solution which is not part of the statement of the lemma. Hence it is confusing the reader. We will make appropriate changes here.

7. **Page 11, proof of Lemma 4.5. This proof needs some polishing. I believe one can directly appeal to Schur complements, and the first two lines are just confusing. Besides, there is a typo on the inverse of $A : a_{22} = a$, and not $d$.**
   Thanks for pointing this out. We will reword the first couple sentences, and fix the typo accordingly. On the high level, this is supposed to be a fairly elementary (but maybe a bit tedious) technical result.

8. **The paper could have provided a more intuitive explanation of the joint regularization used.**
   We agree that the current approach appears more in a bottom-up way, i.e. starting from the desired algebraic properties of area-convexity and engineering an appropriate function. A more intuitive interpretation of the joint regularization would offer further insight to extend the result. On a high level, comparing to independent regularization (i.e. using $\phi_1(x) + \phi_2(y) + \phi_3(z)$ to regularize), the joint regularization can be viewed as adaptively using a different regularization on $x$ based on the current $y, z$, which makes the regularization behaves more locally. Although this local nature makes the guarantees weaker, it is sufficient since we only need such guarantees within the local neighborhood where the update step is carried out. This is also the key to why $\rho$ is $O(n^{o(1)})$. Intuitively, we weigh the term involving $x_i$ in the regularization dynamically based on how 'significant' each $x_i$ is at the current step, and the total 'significance' over all variables at any time can be bounded by the infinity norm rather than the number of coordinates $n$. In passing, we would like to remark that this is still an algebraic interpretation and obtaining a geometric interpretation might be difficult because of high dimensionality of the problem. If one would really like to go in this direction then a starting possibility would be the 2-D toy example in line 183. However, we haven't ventured into this side yet.

[Meta-Review · NeurIPS 2019]

This paper shows that Mixed Packing Covering Linear programs can be eps-approximated in O(1/eps) iterations. This work builds on Sherman's approach to max flow. This is a strong result on an important class of problems. Optimization problems of this nature arise frequently in machine learning applications and reducing the eps dependence from quadratic to linear is a significant improvement. The reviewers liked the result and the presentation and strongly supported acceptance. One aspect that could be improved is the motivation. While optimization problem are central to ML, mentioning some applications of these problems to ML applications (e.g. to optimal transport) would make the connection to this conference more explicit. While the paper is well-written the authors are encouraged to remind the general Neurips audience of the significance of these results to ML applications.